# Extraction and Spatio-Temporal Analysis of Impervious Surfaces over Dongying Based on Landsat Data

**Jiaqi Shen** [1], **Yanmin Shuai** [2,3,4,5,*], **Peixian Li** [1], **Yuxi Cao** [1] and **Xianwei Ma** [3]

1   College of Geoscience and Surveying Engineering, China University of Mining and Technology (Beijing), Beijing 100083, China; jiaqi_sh@student.cumtb.edu.cn (J.S.); lipx@cumtb.edu.cn (P.L.); cyxcumtb@student.cumtb.edu.cn (Y.C.)
2   Xinjiang Institute of Ecology and Geography, Chinese Academy of Sciences, Urumqi 830011, China
3   College of Surveying and Mapping and Geographical Sciences, Liaoning Technical University, Fuxin 123000, China; 471820581@std.lntu.edu.cn
4   Research Centre for Ecology and Environment of Central Asia, Urumqi 830011, China
5   University of Chinese Academy of Sciences, Beijing 100049, China
*   Correspondence: shuaiym@ms.xjb.ac.cn

**Abstract:** It is necessary to understand the relationship between the impervious surface area (ISA) distribution, variation trends and potential driving forces over Dongying, Shandong Province. We extracted ISA information from Landsat images with 3–5 year intervals during 1995 to 2018 using Minimum Noise Fraction (MNF) transform, Pixel Purity Index (PPI), and Linear Spectral Mixture Analysis (LSMA), followed by the analysis on three driving forces of ISA expansion (physical geography, socioeconomic factors, and urban cultural features). Our results show the retrieved ISA thematic map fit the limited requirement of root mean square error (RMSE). The correct classification accuracy of ISA is greater than 83.08%. Further, the cross–comparison exhibits the general consistent with the ISA distribution of the land use classification map published by the National Basic Geographic Information Center. The gradual increasing trend can be captured on the expansion of ISA from 1995 to 2018. Despite of the central region always shown as the high ISA density, it still keeps increasing annually and radiating the surrounding region, especially in the southward which has formed into a new large–scale and high intensity of ISA in 2015–2018. Though the ISA patches scattered in the west region or along the northern and eastern part of the ocean coastline are still small, the expansion trend of ISA can be detected. The expansion intensity index (EII) of ISA measuring the situation of its expansion changes from the lowest value 0.12% between 1995 and 2000 up to the highest 0.73% between 2000 and 2005. Richly endowed by nature, the city's natural geographical environment provides an elevated chance of further urbanization. The rapid increase of regional economy provides a fundamental driving force for expanding ISAs. The development of urban culture promotes the sustainable development of ISAs. Our results provide a scientific basis for future urban land use management, construction planning, and environmental protection in Dongying.

**Keywords:** impervious surface; mixed pixel; linear spectral mixing analysis; endmember

## 1. Introduction

Rapid urbanization has led to the replacement of a large amount of natural land (e.g., grassland and forest areas) by residential, transportational, industrial, and commercial land, which is mostly composed of impervious surface areas (ISAs) [1–3]. ISAs are ground surfaces through which water cannot penetrate, such as surfaces covered by buildings, hardened pavement, and stadiums [4]. As the impervious surface is closely related to commercial, industrial, and residential areas, it has been widely applied as an important indicator of land use/land cover transformation from natural features to urban features [5–7]. From an urban hydrology perspective, the increasing coverage of impervious surfaces increases the speed and volume of urban surface runoff, greatly increasing the pressure on

municipal drainage and flood control [1]. Recently, successive rainstorms in Jiangsu, Henan, and Shaanxi provinces have caused serious urban ponding, resulting in the trapping of people and vehicles, damage to roads and houses, and threats to peoples' lives and property. Therefore, research on ISA distribution is very important for urban future planning and management. A study [8] of the relationships among land cover change, population growth, road density, and the relative change of flooding areas from 2000 to 2017 showed that an increase in areas with high susceptibility to flooding is related to the regional population growth rate, with the Pearson correlation coefficient calculated to be 0.496. Additionally, increased runoff due to urbanization exacerbates nonpoint source pollution discharge in waterways [9]. This increase in ISAs also weakens the effect of precipitation infiltration, thus affecting groundwater recharge [10]. Additionally, ISAs absorb large amounts of heat, and, therefore, their coverage, distribution, and changes have been associated with surface temperature variations [11,12]. In one study, the impact of urbanization was estimated by comparing observations in cities with those in surrounding rural areas. The authors used the difference between the trend in observed surface temperatures and corresponding trends in the reconstruction of surface temperatures determined through the reanalysis of global weather over the past 50 years to examine the impact of land use changes on surface warming. The results suggested that half of the observed decrease in diurnal temperature range occurred because of urban and other land use changes [13]. Changes in land use and land cover greatly affect the energy balance and biogeochemical cycles and, thus, affect land surface properties and the ecosystem environment. A study [14] of 35 years of satellite data provided a comprehensive record of global land change dynamics from 1982 to 2016, showing that 60% of land changes are associated with direct human activities and 40% with indirect drivers such as climate change. Land use change exhibits regional dominance, including urbanization, tropical deforestation, and farmland intensification. The impervious layer may also have a direct or indirect influence on many environmental factors, and changes in ISA ratios often directly coincide with urban development and expansion. Therefore, estimating and monitoring the distribution, development, and change in ISAs in urban areas have recently gained attention [15–18], particularly in developed countries [19]. The United States Geological Survey (USGS) has included impervious surface data in the second edition of its National Land Cover Database (NLCD), and, subsequently, has included impervious layer distribution survey data and research in the NLCD databases in 2006 and 2011 [20]. The official impervious layer distribution monitoring information collected by the USGS has provided reliable support for studying urban and regional development and expansion [21–24]. In particular, these datasets have been instrumental in characterizing ISA distribution and expansion, thus contributing to the development of management strategies for urban construction and environmental protection. Therefore, it is important to map the ISA evolution characteristics and better understand the urban process to guide the sustainable development of cities.

Traditionally, supervised and unsupervised classification methods were widely used in impervious surfaces' estimation. Although these methods are easy to implement, they only consider one land type in a pixel and ignore spatial heterogeneity. In fact, there are a large number of mixed pixels in an image containing more than one type of land. This can lead to information loss and accuracy reduction when using traditional methods [25]. Therefore, many approaches have been developed for the mapping of impervious surfaces at the subpixel level, such as decision tree, regression tree, artificial neural network, and Linear Spectral Mixture Analysis (LSMA) [11,26,27]. Among these, LSMA is the most widely used for solving the spectral mixing problem, as it shows better performance than other methods in many fields, such as land cover classification [28], wetland classification [29], and urban classification [30]. In addition, numerous studies have been conducted to improve the performance of Spectral Mixture Analysis (SMA) [31]. Ridd proposed a Vegetation–Impervious Surface–Soil (VIS) model [32] to enable the extraction of ISAs from complex urban landscapes. This model eliminates easily distinguishable water features and uses a linear combination of vegetation, ISAs, and soil to simulate urban land cover.

Phinn et al. [33] used pixel spectral analysis to estimate ISA distribution. Wu et al. used a linear spectral mixing model to demonstrate that the ISA can be obtained from the sum of high and low albedo components [2]. Yuan et al. used ISA data to extract impervious ground ratios in Minnesota [11]. Moreover, Yang [34] used Landsat images to analyze the ISA spatial distribution in the Nanjing urban area using three methods: linear spectral mixed decomposition with limited conditions, negative correlation model of vegetation cover and ISA, and supervised classification. Importantly, the author demonstrated that the linear spectral mixed decomposition method had the best performance. Lu and Wu characterized urban land cover types using Thematic Mapper (TM) images via the linear combination of four spectral endmembers: high albedo, low albedo, vegetation, and soil. Next, the distribution of ISA in the Beijing urban area was estimated using a linear spectral mixture model with limited conditions [35]. Xia used the linear spectral mixing model to extract the impervious layer ratio of two temporal phases in the downtown area of Xuzhou, at a subpixel scale, through the decomposition of mixed pixels, and analyzed the impervious layer information extracted from two temporal hyperspectral remote sensing images [36].

Although LSMA is widely used for ISAs extraction because of its robust theoretical basis, high efficiency, and precision, for different regions, the spectral complexity and land types, and the applicability of LSMA differ. Therefore, we applied LSMA to Dongying City, which is surrounded by the sea in the east and north and has complex land types. We used LSMA to extract the ISA of Dongying and generate an ISA distribution map with 30 m resolution over a long time series from 1995 to 2018 based on Landsat. Landsat's on orbit and historical data opening policy provides data conditions for the ISA distribution of long time series. Obtaining the long-term ISA distribution and summarizing the temporal and spatial expansion law of the ISA may provide guidance for policymakers and city planners in sustainable land use and infrastructure planning. However, using LSMA to extract the ISA does not explain the driving force of ISA expansion [37]. Therefore, we performed long-term (1995–2018) monitoring of Dongying urbanization, discuss the changing trend in the ISA, and provide insight into the urbanization process of Dongying. Based on remote sensing image preprocessing, water mask, Minimum Noise Fraction (MNF) transform, and Pixel Purity Index (PPI) calculation, LSMA is used to extract the ISA. A qualitative evaluation method was used to analyze the relationship between the ISA change trend and three driving forces: physical geography, socioeconomic factors, and urban culture characteristics, so as to clarify the mechanisms driving ISA changes in Dongying.

## 2. Materials and Methods

### 2.1. Study Area

Dongying City is a central city in the Yellow River Delta in Shandong Province, China, located at 118°07′ E–119°10′ E and 36°55′ N–38°10′ N, as shown in Figure 1. Dongying city has a warm, temperate, continental monsoon climate and its terrain slopes from southwest to northeast along the Yellow River. Dongying city is an important node of Bohai Rim Economic Zone and an important part of the Shandong Peninsula urban agglomeration. This city is in a pivotal position connecting the Central Plains Economic Zone and Northeast Economic Zone, Beijing Tianjin Tangshan Economic Zone and Jiaodong Peninsula Economic Zone.

By 2019, the city had jurisdiction over three districts and two counties, with a total area of 8243 km$^2$, a permanent population of 2,179,700, an urban population of 1,509,300, and an urbanization rate of 69.24% [38]. In August 2019, China Customs magazine sponsored by the General Administration of Customs of China announced the ranking of "top 100 cities of China's foreign trade" in 2018, with Dongying ranking 31st. In July 2020, the National Patriotic Health Association confirmed Dongying as a national health city in 2019 [39]. In 2020, the GDP of Dongying City was CNY 298.119 billion, an increase of 3.8% from the previous year [40].

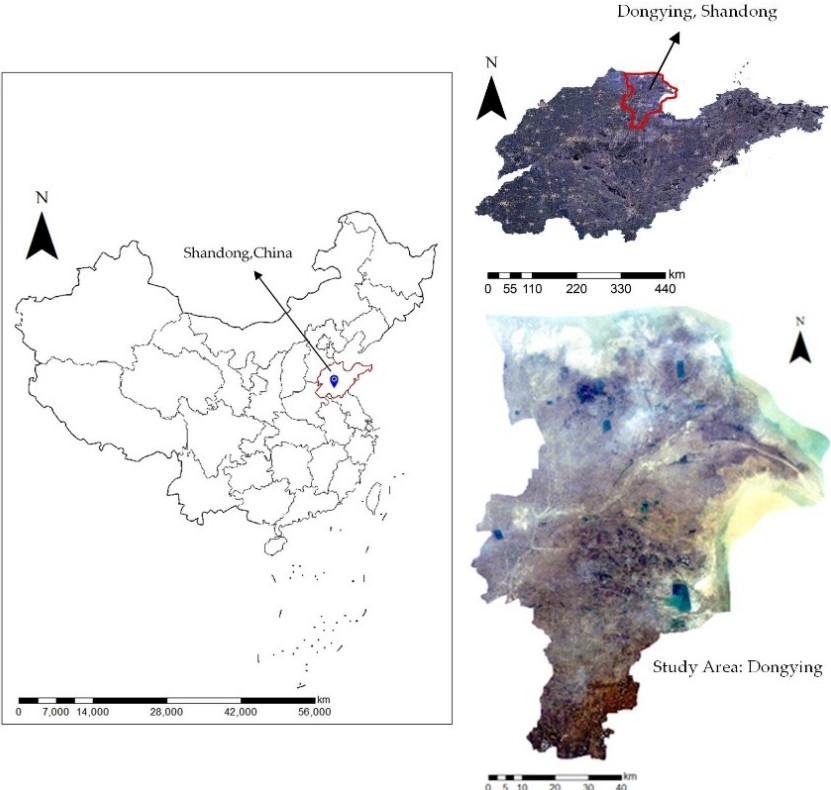

**Figure 1.** Study area: Dongying City, Shandong Province, China.

### 2.2. Data Collection and Preprocessing

Multitemporal Landsat TM and Landsat Operational Land Imager (OLI) images were used in the study [41]. The TM images were acquired on 22 February 1995, 4 February 2000, 1 February 2005, and 14 January 2010. OLI images were acquired on 13 February 2015 and 20 January 2018. The 30 m time series Landsat images from Path 121/Row 34 were selected for this study. The cloud cover of all images was less than 5%, and all images contained six effective bands, including bands 1–5 and band 7, of the TM data, as well as bands 2–7 of the OLI data. Data preprocessing included image sub-setting, as shown in Figure 1, radiometric calibration, and atmospheric correction.

Radiometric calibration is a process involving sensor error elimination to accurately determine the radiation sensor entrance value. A calibration coefficient is used to convert the gray value of an image or Digital Number (DN) value into a radiation brightness value, as shown in Equation (1).

$$L_\lambda = DN \times G + B \tag{1}$$

where $L_\lambda$ is the measured spectral radiance, $DN$ is the recorded electrical signal value, $G$ is the sensor gain, and $B$ is the sensor bias. A radiometric calibration module is then used to calibrate the clipped image, which can automatically read the information in the remote sensing image, including the sensor type, imaging time, and solar altitude angle. The calibration type is radiance and is the data saved in band interleaved by line (BIL) format, which can be used for atmospheric correction.

The energy captured by Landsat sensors is influenced by the Earth's atmosphere. These effects include scattering and absorption due to interactions of electromagnetic radiation with atmospheric particles (i.e., gases, water vapor, and aerosols) [42]. Atmospheric correction aims to determine the true surface reflectance values by removing atmospheric effects resulting from the scattering and absorption of electromagnetic radiation by gases and aerosols when passing through the atmosphere to the satellite sensor [43]. In Landsat TM and OLI data, the dominant atmospheric effect is scattering, which is additive to the remotely sensed signals, whereas the multiplicative effect from absorption is often ne-

glected because the TM and OLI bands are selected to avoid effects due to absorption [44]. Under certain conditions, calibration of image data to radiance units is necessary prior to classification and change detection using multitemporal images [45]. Considering the effect of the atmosphere can avoid improper interpretation of images [46].

The Fast Line-of-sight Atmospheric Analysis of Spectral Hypercubes (FLAASH) atmospheric correction model was used to calibrate the radiometric images. Based on the solar spectral range (excluding thermal radiation) and flat Lambert body (or approximate plane Lambert body), the spectral radiance formula of the pixel received by FLAASH at the sensor is shown in Equation (2).

$$L = \left( \frac{A * \rho}{1 - \rho_e * S} \right) + \left( \frac{B * \rho}{1 - \rho_e * S} \right) + (L_a) \tag{2}$$

where $L$ is the total radiance received by the pixel at the sensor; $\rho$ is pixel surface reflectivity; $\rho_e$ is the average surface reflectance around the pixel; $S$ is the atmospheric spherical albedo; $L_a$ is the atmospheric backscattering emissivity (atmospheric radiation); $A$ and $B$ are two coefficients that depend on atmospheric and geometric conditions.

Equation (2) can be divided into three parts (with brackets as the dividing line). The first part is the radiation intensity of solar radiation entering the sensor directly after entering the earth's surface through the atmosphere; the second part is radiation entering the sensor after atmospheric scattering; the third part is the atmospheric backscattering rate (atmospheric radiation). Central longitude and latitude, sensor type, sensor altitude, ground elevation, pixel size, and flight date can be obtained from the image header file. The mid-latitude winter atmospheric mode is selected based on season–latitude information of the images. The rural aerosol model is selected based on the images feature. The relevant parameter settings of atmospheric correction are shown in Table 1. The image after atmospheric correction, take the 2015 OLI image as an example, is shown in Figure 2.

### 2.3. Mapping Impervious Surface Area (ISA) Distribution

The data used herein includes the Dongying administrative boundaries, Landsat TM images, and OLI images. Data preprocessing and water masking were performed first. Next, most of the noise was filtered out through MNF, and the purest pixel in the image was identified through PPI calculation. Finally, ISAs were extracted via LSMA. According to the calculation results, the area, expansion ratio, and expansion intensity index (EII) of ISA in Dongying from 1995 to 2018 were calculated. Then, the qualitative evaluation method was used to analyze the relationship between ISA change trend and three driving forces: physical geography, socioeconomic factors, and urban cultural feature, so as to clarify the driving mechanism of ISA change in Dongying. Figure 3 illustrates a flowchart of our procedures.

**Table 1.** Parameter setting of atmospheric correction.

| Image Sequence Number | 1 | 2 | 3 | 4 | 5 | 6 |
|---|---|---|---|---|---|---|
| Central Longitude | | | 118°54′51.75″ | | | |
| Central Latitude | | | 37°28′16.25″ | | | |
| Sensor Type | | Landsat TM5 | | | Landsat-8 OLI | |
| Sensor Altitude | | | 705 km | | | |
| Ground Elevation | | | 8.8 m | | | |
| Pixel Size | | | 30 m | | | |
| Flight Date | 22 February 1995 | 4 February 2000 | 1 February 2005 | 14 January 2010 | 13 February 2015 | 20 January 2018 |
| Atmospheric Models | | | Mid-Latitude Winter | | | |
| Aerosol Model | | | Rural | | | |

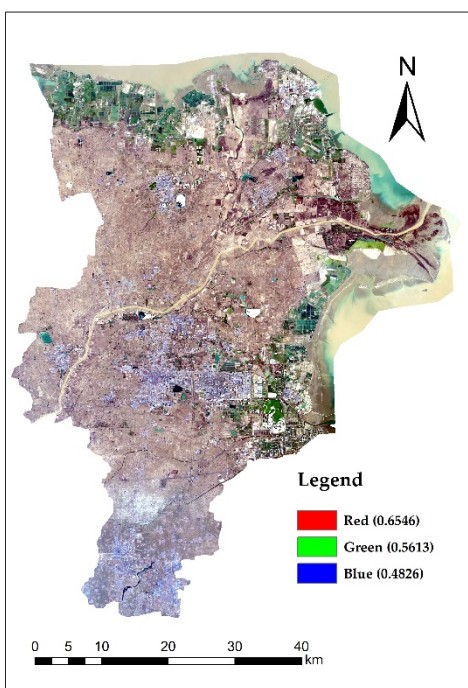

**Figure 2.** Dongying image after atmospheric correction by FLAASH module.

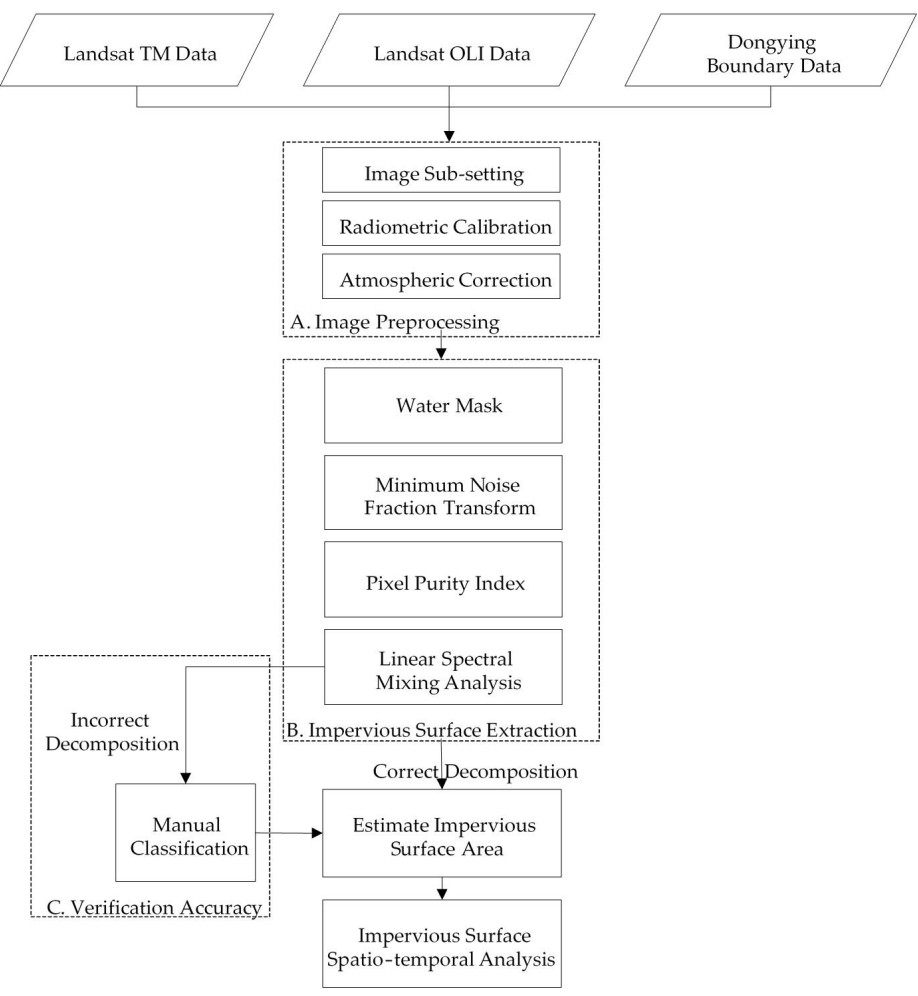

**Figure 3.** Flowchart for extraction and spatio-temporal analysis of ISA of Dongying from 1995 to 2018 based on Landsat data.

### 2.3.1. Water Mask

First, water bodies should be removed from the images to avoid confusing the classification between water bodies and impervious surfaces [47]. Because of its simplicity and accuracy, the Modified Normalized Difference Water Index (MNDWI) is widely used to extract water bodies. MNDWI uses a specific band of a remote sensing image for normalized difference processing to highlight water information in the image [48]. The calculation method is shown in Equation (3). After water body masking treatment, the water area is eliminated, reducing the influence of the water area on ISA extraction.

$$\text{MNDWI} = \frac{\text{Green} - \text{MIR}}{\text{Green} + \text{MIR}} \tag{3}$$

where, MIR is the mid-infrared band (i.e., band 5 in the TM image and band 6 in the OLI image). Green is the green light band, which corresponds to band 2 in the TM image and band 3 in the OLI image.

After calculating the MNDWI, the boundary value between water and non-water was obtained by density segmentation, and the water body was extracted using a threshold. The image obtained after water body masking, take the 2015 OLI image as an example, is shown in Figure 4.

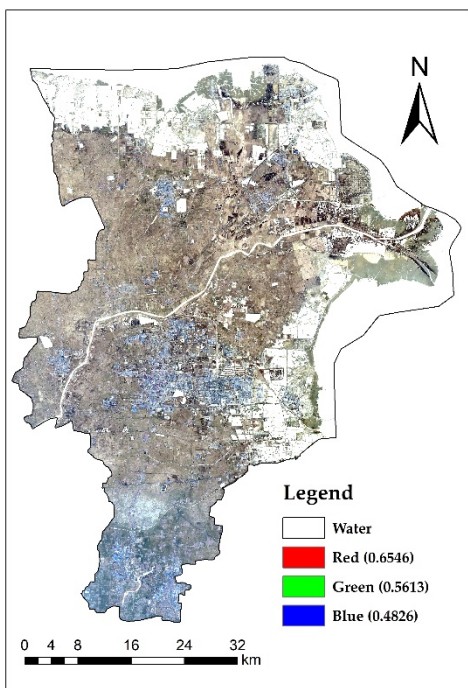

**Figure 4.** Dongying true color image after water masking with MNDWI. Within the boundary of Dongying vector, the white area is water, which has the same value as the background, and the non-water area is stretched according to the red, green, and blue band.

### 2.3.2. Minimum Noise Fraction (MNF)

The noise of remote sensing images and correlation between spectral bands reduce the number of data features that can be extracted from the images [49]. To resolve this, MNF transform can be used for dimension reduction and noise isolation of multispectral data, and to reduce between-band correlation and the number of subsequent calculations [50]. MNF transform is essentially a double–stacked principal component analysis. MNF transform can separate noise from effective information by arranging components according to the signal–to–noise ratio. The first transform is based on estimation of the noise covariance matrix to adjust and separate the data noise. The transform noise data represents the noncorrelation between bands, which is often referred to as noise whitening. In the second

transform, standard principal component transform is applied to the noise whitening data, after which the final eigenvalues and internal dimensions of the related image data are analyzed.

Using OLI image data from 13 February 2015 as an example, the first three bands of the MNF transform account for 91.19% of the total remote sensing image information. Therefore, the first three principal components can be used to represent the basic data dimensions. Table 2 summarizes the MNF transform results. The eigenvalue after MNF transform represents the importance of each component, and the eigenvalue of each component decreases steadily as the number of MNF components increases. Components with higher eigenvalues can be used to express image information [51].

**Table 2.** Eigenvalues and specific gravity of each MNF component.

| MNF | Eigenvalue | Cumulative Percentage |
|---|---|---|
| 1 | 209.1497 | 67.00% |
| 2 | 57.9419 | 85.56% |
| 3 | 17.5451 | 91.19% |
| 4 | 14.0268 | 95.68% |
| 5 | 8.1962 | 98.30% |
| 6 | 5.2913 | 100.00% |

Using TM image data from 22 February 1995 as an example, we compared the signal-to-noise ratio of each component in the same region of the MNF image, as shown in Figure 5. Notably, the first three components exhibited less image noise and were generally clearer.

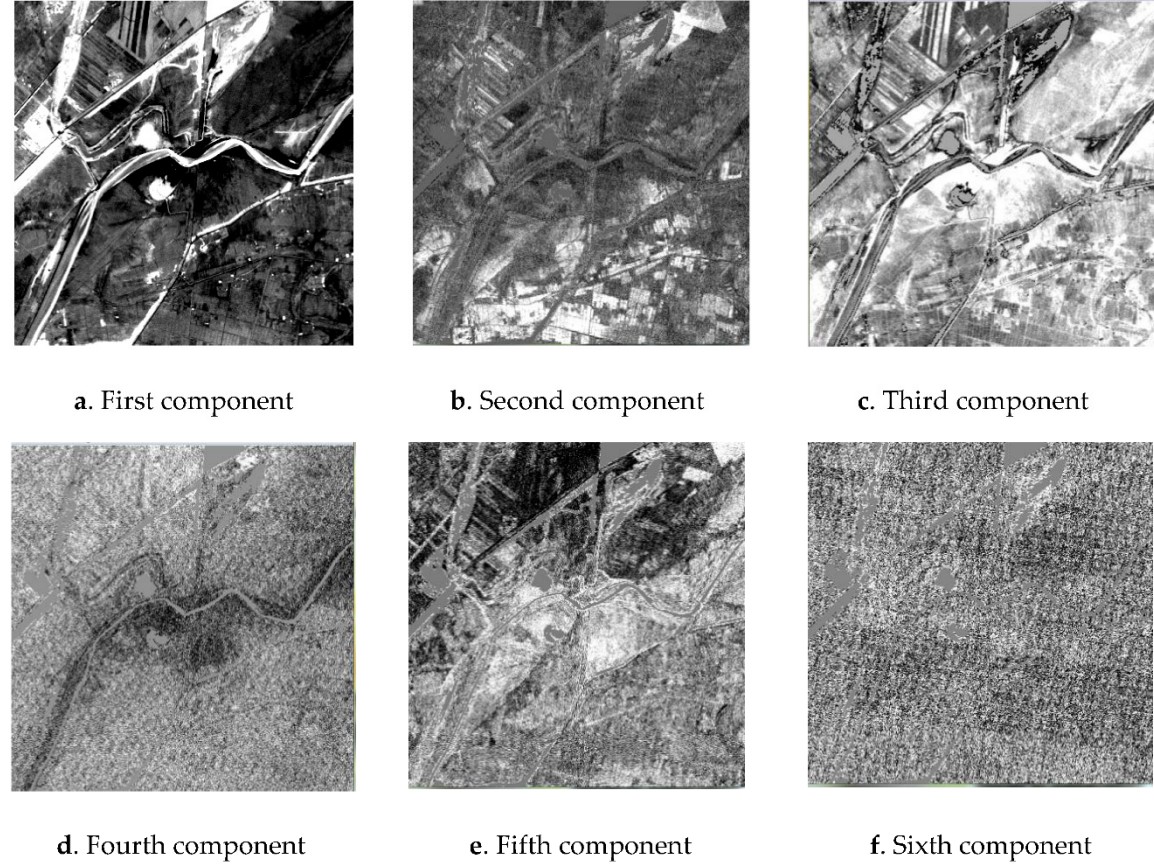

**a**. First component          **b**. Second component          **c**. Third component

**d**. Fourth component          **e**. Fifth component          **f**. Sixth component

**Figure 5.** Six components of Minimum Noise Fraction (MNF) images derived for the same area of Dongying City.

### 2.3.3. Pixel Purity Index (PPI)

PPI calculation [52] is a method used to identify the purest spectrum pixel in multi-spectral or hyperspectral images. PPI treats each pixel as an n–dimensional vector, and all pixels form a vector space. In this vector space, there must be a basis for all vectors located on the boundary, and all space vectors can be represented by linear combinations of these bases. The probability of appearing on the edge of random unit vectors is the largest when these vectors at the boundary position are projected onto the random unit vectors, which can be expressed as the purity index. Mixed pixels surrounded by boundary pixels can be decomposed through linear combinations of pure pixels. After PPI calculation, the DN value indicates the number of times that the pixel was marked as a pure pixel.

An MNF image was used as an input data for PPI calculation. To meet the endmember extraction requirements, 10,000 PPI iterations were established. The threshold factor refers to data bits as an index. For example, if the threshold value is 2, only pixels with a difference between the DN value and extremum pixel of more than two digits is labeled as an extremum. The threshold selects pixels at the end of the mapped pixels. The threshold should optimally be 2–3 fold the noise level [53]. The noise level is considered as the noise pixel. To distinguish between normal pixels and noise, the median value of adjacent pixels was calculated. If the values were greater than a certain threshold, the pixels were marked as noise.

### 2.3.4. Extract Endmembers

In the process of linear spectral mixing decomposition, the choice of endmembers directly determines the accuracy of the final mixed pixel decomposition result. The selection of endmembers depends on the specific characteristics of the study area. Based on the complex types of ground features and intensive construction land, the high albedo–low albedo–vegetation–soil mixed pixel decomposition model was selected for ISA extraction in Dongying. In this model, the high albedo end elements mainly include high light roof materials, glass, and ceramic tiles with a high reflective effect. Low albedo end elements mainly include asphalt pavement and carbon steel material. Vegetation end elements include woodlands, urban inner green lands, and farmland. Soil endmembers are mainly bare soil. Water bodies do not affect the selection of endmembers as the water has been masked in the previous step. Four types of ground objects (i.e., high albedo, low albedo, vegetation, and soil) were selected based on the original image, Google Earth images, PPI results, and distribution characteristics of four endmembers in the first three MNF components.

### 2.3.5. Linear Spectral Mixture Analysis (LSMA)

LSMA assumes that there is no interaction between light endmembers. The mixed pixel spectrum is regarded as a linear combination of all components of the corresponding spectral values according to its area proportion [54,55]. Equations (4)–(5) detail the LSMA mathematical model.

$$x_b = \sum_{k=1}^{n} s_k a_{kb} + e_b \tag{4}$$

$$\sum_{k=1}^{n} s_k = 1, 0 \leq s_k \leq 1 \tag{5}$$

where $x_b$ is the reflectivity of band $b$; $s_k$ is the proportion of the area occupied by $k$; $a_{kb}$ is the reflectivity of $k$ in band $b$; $b$ is the number of spectral bands, $b = 1, 2 \ldots m$, $m$ is the total number of spectral bands in the image; $k$ is the $k^{th}$ end element, $k = 1, 2, 3,$ and 4; $n$ is the total number of endmembers in the image. Given that $x_b$ is known, if $a_{kb}$ is obtained again, it is feasible to solve $s_k$ via the least square error method. Therefore, resolving the endmember spectrum is key to implementing the linear spectral model.

The correctness of LSMA model can be determined by checking root mean square error (RMSE) of the residual $e_b$ of each band in the image, as shown in Equation (6).

$$\text{RMSE} = \sqrt{\frac{\sum_{b=1}^{m}(e_b)^2}{m}} \tag{6}$$

where $e_b$ is the residual; RMSE is the root mean square error.

## 3. Results

### 3.1. Impervious Surface Mapping

The linear spectral mixture decomposition maps of land types in Dongying in 1995, 2000, 2005, 2010, 2015, and 2018 were obtained with LSMA based on the collected endmember model, and then six periods of ISA distribution images were obtained, as shown in Figure 6.

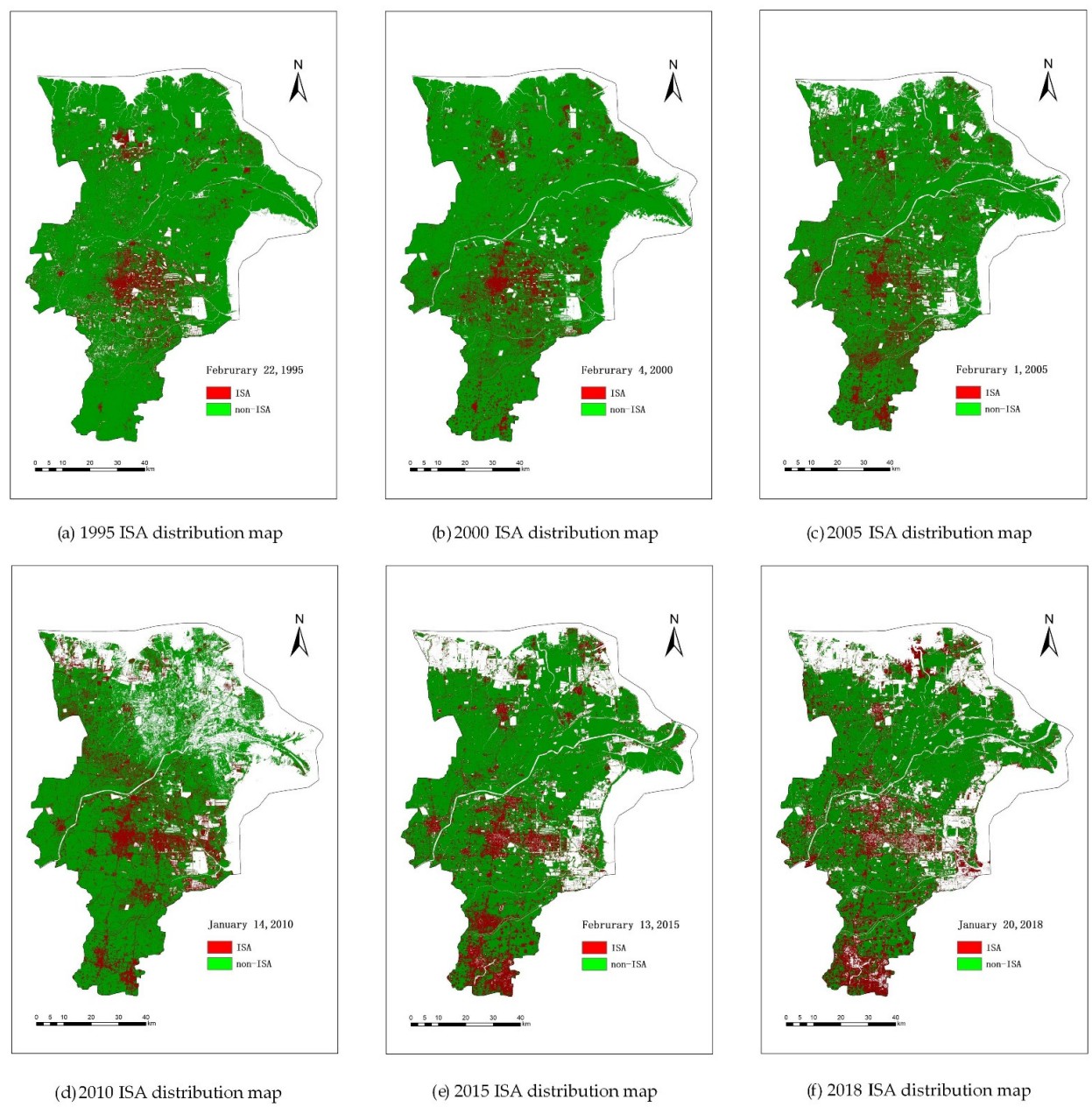

**Figure 6.** Comparison of ISA distributions at temporal scale showing the ISA expansion patterns in Dongying metropolis, 1995–2018.

As shown in Figure 6, ISAs gradually increased from 1995 to 2018. The scope of the ISA was the smallest in 1995 but it was large in 2015 and 2018. In terms of the regional distribution, the central region has always been a region with high ISA density, and its ISA is still increasing year by year and radiating around; ISA expansion is more obvious in the south, which had formed a new large–scale and highly intensive area in 2015 and 2018. The ISA values in the west, and in the north and east near the ocean, are relatively low, but there are also expansion phenomena. ISA can be estimated according to the ISA abundance map in Figure 6. Figure 7 is a line chart of Dongying's ISA from 1995 to 2018. The figure clearly shows that the ISA increased consistently each year, with the largest increase from 2000 to 2005.

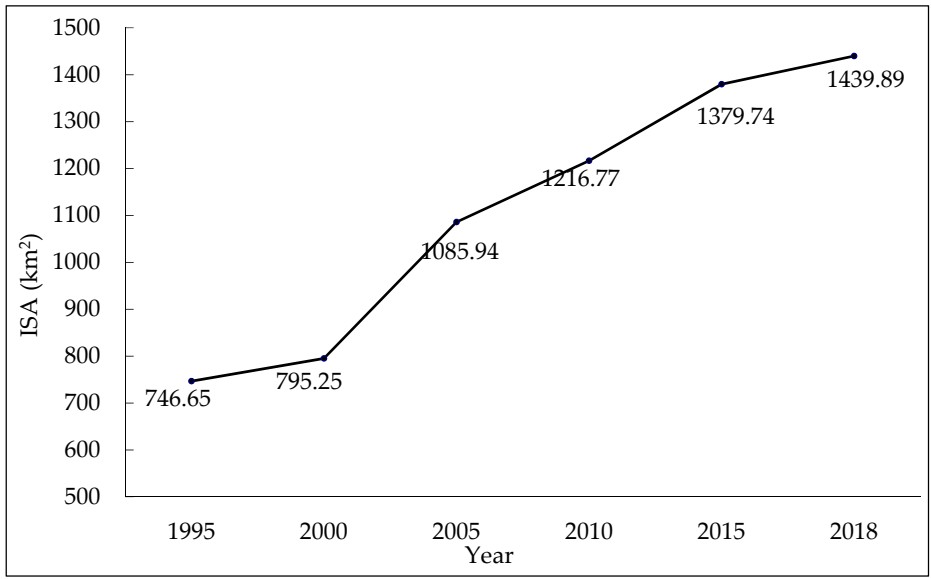

**Figure 7.** Line chart of ISA change trend in Dongying City from 1995 to 2018.

ISA ratio is used to evaluate the expansion speed of ISA in different periods. It is obtained by dividing the difference between the ISA at the beginning and end of the study by the time interval. The ISA ratio of Dongying from 1995 to 2018 is shown in Table 3.

**Table 3.** Dongying ISA ratio from 1995 to 2018.

| Year | 1995 | 2000 | 2005 | 2010 | 2015 | 2018 |
|---|---|---|---|---|---|---|
| ISA ratio (%) | 9.42 | 10.04 | 13.71 | 15.36 | 16.74 | 17.47 |

As shown in Table 3, the ISA ratio increased yearly and showed variable change characteristics. The ISA ratio was lowest in 1995 but increased in the following years. The increase in the ISA ratio was largest between 2000 and 2005. In 2018, the ISA ratio reached 17.47%, which was the maximum ISA ratio in the study period.

ISA EII refers to the proportion of ISA expansion in the study area during the study period. This value is used to quantitatively compare the degree of ISA expansion and measure the strength and speed of ISA expansion in different periods. Equation (7) is the EII calculation formula.

$$\text{EII} = \frac{S_b - S_a}{TLA} \times \frac{1}{T} \times 100\% \tag{7}$$

where EII is the expansion intensity index, $S_a$ and $S_b$ are the ISA at the beginning and end of the study, $T$ is the time interval, and $TLA$ is the total area of the study area.

ISA EII is shown in Table 4. The minimum ISA EII (0.12%) occurred between 1995 and 2000, whereas the highest (0.73%) occurred between 2000 and 2005.

**Table 4.** ISA Expansion Intensity Index from 1995 to 2018.

| Period | 1995–2000 | 2000–2005 | 2005–2010 | 2010–2015 | 2015–2018 |
| --- | --- | --- | --- | --- | --- |
| ISA EII (%) | 0.12 | 0.73 | 0.33 | 0.41 | 0.24 |

### 3.2. Precision Evaluation

First, the ISA classification accuracy was verified by RMSE. The RMSE image effectively reflects the accuracy of the decomposition result and can be directly used to evaluate the decomposition result [56]. More highlighted regions in the RMSE diagram result in a larger RMSE value, indicating a poorer selection of endmembers and decomposition effect and vice versa. Previous studies demonstrated that the average RMSE must be less than 0.02 to ensure the decomposition effect [2]. Using data from 2015 as an example, the spatial distribution of RMSE is shown in Figure 8, and a histogram is shown in Figure 9. The RMSE value of most pixels is 0 (i.e., less than the allowable value 0.02), meeting the accuracy requirements. This result confirmed that the selected end element was accurate, and that the decomposition result was reliable.

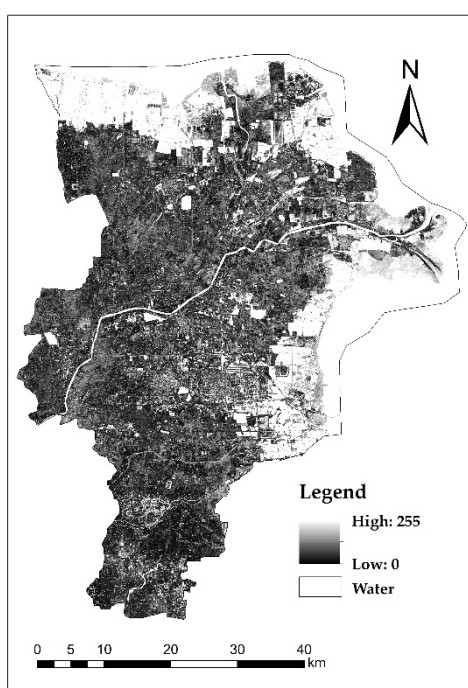

**Figure 8.** RMSE spatial distribution of the linear spectral decomposition.

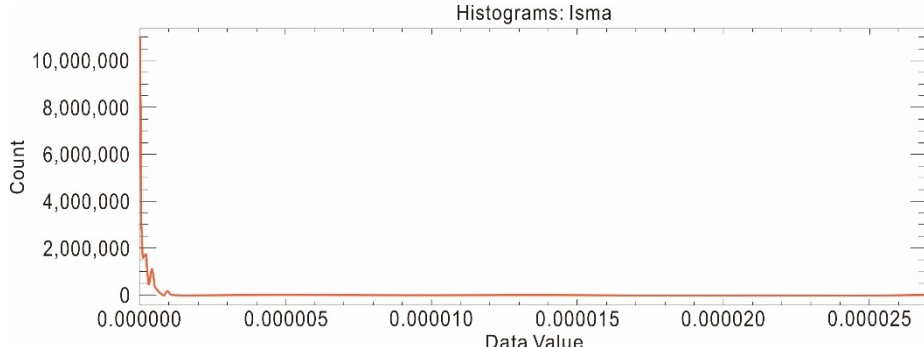

**Figure 9.** RMSE Histogram for spectral decomposition: the RMSE value of most pixels is close to 0, which is much less than 0.02, meeting the decomposition requirements.

Second, samples were randomly selected on the original image for testing. As there were no high spatial resolution images with long time series, verification samples could only be selected from original images. The validation samples ensured that there was no overlap with the classification samples and that the categories were specific and could be interpreted accurately. The confusion or error matrix plays a central role in meeting both the accuracy assessment and area estimation objectives [57]. Therefore, the confusion matrix was used for accuracy evaluation, and indicators related to ISA classification are shown in Table 5.

**Table 5.** Indicators related to ISA classification of land classification confusion matrix in Dongying from 1995 to 2018.

| Year | Overall Accuracy | Kappa Coefficient | Selected ISA Pixels | Correctly Classified ISA Pixels | ISA Accuracy |
|------|------------------|-------------------|---------------------|---------------------------------|--------------|
| 1995 | 92.32% | 0.88 | 2651 | 2354 | 88.80% |
| 2000 | 90.81% | 0.87 | 3390 | 3225 | 95.13% |
| 2005 | 91.00% | 0.88 | 3074 | 2554 | 83.08% |
| 2010 | 89.56% | 0.86 | 3726 | 3519 | 94.44% |
| 2015 | 88.90% | 0.85 | 2592 | 2534 | 97.76% |
| 2018 | 87.61% | 0.83 | 2371 | 2311 | 97.47% |

Using the LSMA method to extract ISA, the overall classification accuracy was more than 87.61%, Kappa coefficient was more than 0.83, and the accuracy of ISA correct classification was more than 83.08%, demonstrating that this method can extract ISA effectively. The highest ISA classification accuracy was 97.76% in 2015. These results show that the classification effect is good.

Third, using the 1 km grid data of the land use classification map as an auxiliary reference, the land use classification data was obtained from the global surface coverage data product service website of the National Basic Geographic Information Center (DOI:10.11769), as shown in Figure 10. Figure 9 was compared with Figure 6 to verify whether the overall trend was consistent. To facilitate visual comparison, the construction land in the existing land classification map is shown in red, and all other land types are shown as green, which is consistent with the color in Figure 6.

As shown in Figure 9, the ISA in Dongying gradually expanded from 1995 to 2018. From the perspective of regional distribution, the middle part of Dongying has always been an area with high ISA density, and its ISA is still gradually increasing and radiating around; the ISA expanded most obviously in the south. ISA is less distributed in the west and east, but there are also some expansion phenomena. These phenomena are consistent with the data shown in Figure 6. However, there were differences in the ISA distribution between Figures 6 and 9, mainly in the following points:

a.  From 1995 to 2018, there was a dense ISA distribution area in the northeast of Figure 9, while the ISA distribution in the same position in Figure 6 was relatively thin. By checking the image of Google Earth in the corresponding year, it can be seen that this area includes an oil and gas injection plant, a sea cucumber seedling breeding base, and a salinization farm. Most of these are made up of vegetation and bare soil. The ISA is relatively small and not as dense as shown in Figure 9.

b.  From 2000 to 2015, there was a dense ISA distribution area in the northwest of Figure 9, while the ISA distribution in the same location in Figure 6 was relatively thin. By checking the images of Google Earth in the corresponding years, it can be seen that most of the area is composed of vegetation, cultivated land, and water. The ISA is relatively small and not as dense as shown in Figure 9.

c.  From 1995 to 2018, the northwest area of Figure 6 consists of a dense ISA area, which is relatively thin in Figure 9. By checking the image of Google Earth in the corresponding year, it can be seen that this area is a dense residential area, which is a typical ISA.

d.  From 2005 to 2018, the ISA in the south of Figure 6 has an obvious expansion trend, forming a relatively dense ISA area, while the ISA in the corresponding position of Figure 9 is relatively thin. Viewing the image of Google Earth corresponding to the year, it can be seen that this area is composed of a large number of typical ISAs such as residential areas, factories, hospitals, and schools.

Thus, compared with classified products, the classification results obtained in this paper are more reasonable.

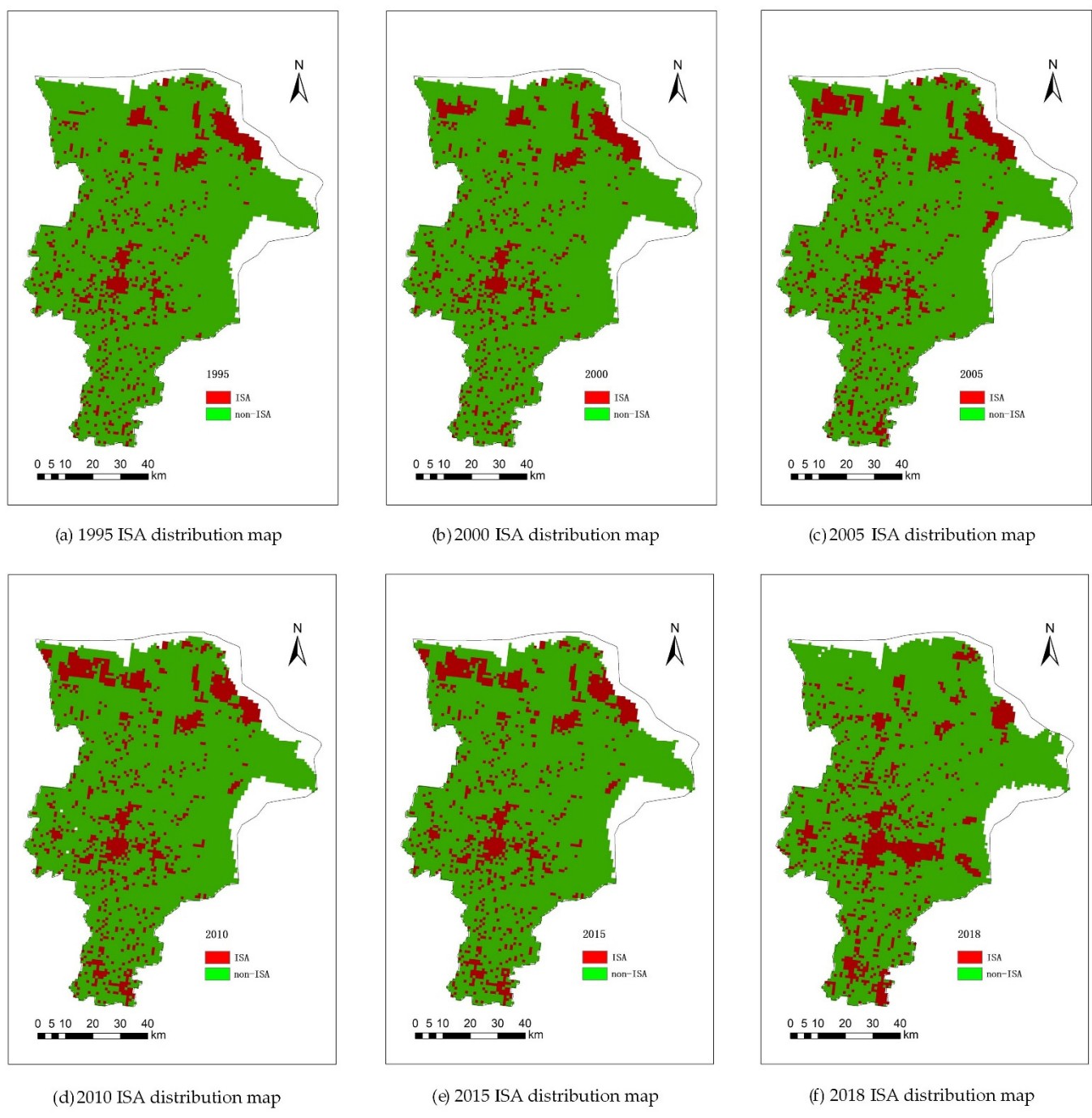

(a) 1995 ISA distribution map    (b) 2000 ISA distribution map    (c) 2005 ISA distribution map

(d) 2010 ISA distribution map    (e) 2015 ISA distribution map    (f) 2018 ISA distribution map

**Figure 10.** Remote sensing monitoring data of land use status in Dongying City from 1995 to 2018.

## 4. Discussion

### 4.1. Spatio-Temporal Analysis of Impervious Surface

#### 4.1.1. Natural Factors on Urban Expansion

Physical geographical conditions are the basic conditions of urban development that determine the speed and direction of urban expansion to some extent and, thus, determine the macro pattern of urban development. Fü et al. [58] proposed that climate change and human activities are the driving forces of land use and land cover change, and that natural conditions are a factor of the driving force. Dongying is in a mid-latitude and features a flat terrain. Therefore, this region exhibits four distinct seasons with cold winters and hot summers. Dongying is rich in land, water, and mineral resources. In 2003, Dongying was rated as a national health city [59]; in 2017, Dongying was selected as the fifth national civilized city and was in the top 200 charming cities with Chinese characteristics in 2017 [60,61]; in 2019, Dongying was rated as a national ecological garden city [62,63]; in 2020, Dongying was included in the list of national double support model cities (counties) [64]. Because of the abundant natural resources, suitable living environment, natural geographical advantages, and superior natural environment in Dongying city, development and opening of the city has occurred, promoting the urbanization process and further promoting ISA expansion.

#### 4.1.2. Socioeconomic Factors on Urban Expansion

Social and economic development are the main driving forces of urban expansion, which is the specific performance of social and economic development at the spatial material level [65]. Based on the actual situation of the study area, data should be used to select indicators according to local conditions, and strive to analyze and evaluate the impact of the economy on ISA. Therefore, seven representative socioeconomic indicators that have been closely linked to ISA expansion were collected from the Dongying yearbook. These included the investment in fixed assets (IIFA), investment in real estate development (IRED), urban per capita disposable income (UPDI), urban per capita consumption (UPCC), primary industry gross domestic product (PGDP), secondary industry gross domestic product (SGDP), and tertiary industry gross domestic product (TGDP), as shown in Table 6.

**Table 6.** Dongying socioeconomic indicators from 1995 to 2018.

| Indicators | IIFA ($\times 10^{12}$ CNY) | IRED ($\times 10^{12}$ CNY) | UPDI ($\times 10^{5}$ CNY/year) | UPCC ($\times 10^{3}$ CNY/year) | PGDP ($\times 10^{9}$ CNY) | SGDP ($\times 10^{9}$ CNY) | TGDP ($\times 10^{9}$ CNY) |
|---|---|---|---|---|---|---|---|
| 1995 | 11.37 | — | 6.17 | 4.17 | 2.83 | 16.25 | 2.12 |
| 2000 | 19.75 | 4.02 | 8.60 | 7.00 | 3.01 | 31.24 | 5.62 |
| 2005 | 60.57 | 4.70 | 14.94 | 9.63 | 4.60 | 66.36 | 16.37 |
| 2010 | 134.90 | 10.03 | 23.80 | 14.74 | 8.23 | 113.68 | 41.96 |
| 2015 | 308.47 | 19.39 | 38.74 | 23.14 | 13.31 | 149.34 | 77.03 |
| 2018 | 255.75 | 16.72 | 47.91 | 28.90 | 14.65 | 162.71 | 101.12 |

As shown in Table 6, all seven socioeconomic indicators increased at each indicator from 1995 to 2018. As illustrated in Figure 11, seven socioeconomic indicators were positively correlated with ISA expansion. The expansion trend of the ISA is generally consistent with the growth trend of socioeconomic indicators. An interaction exists between socioeconomic development and ISA expansion: the rapid and steady development of the social economy improves the speed of the urbanization process, and ISA expansion provides space and a platform for economic development.

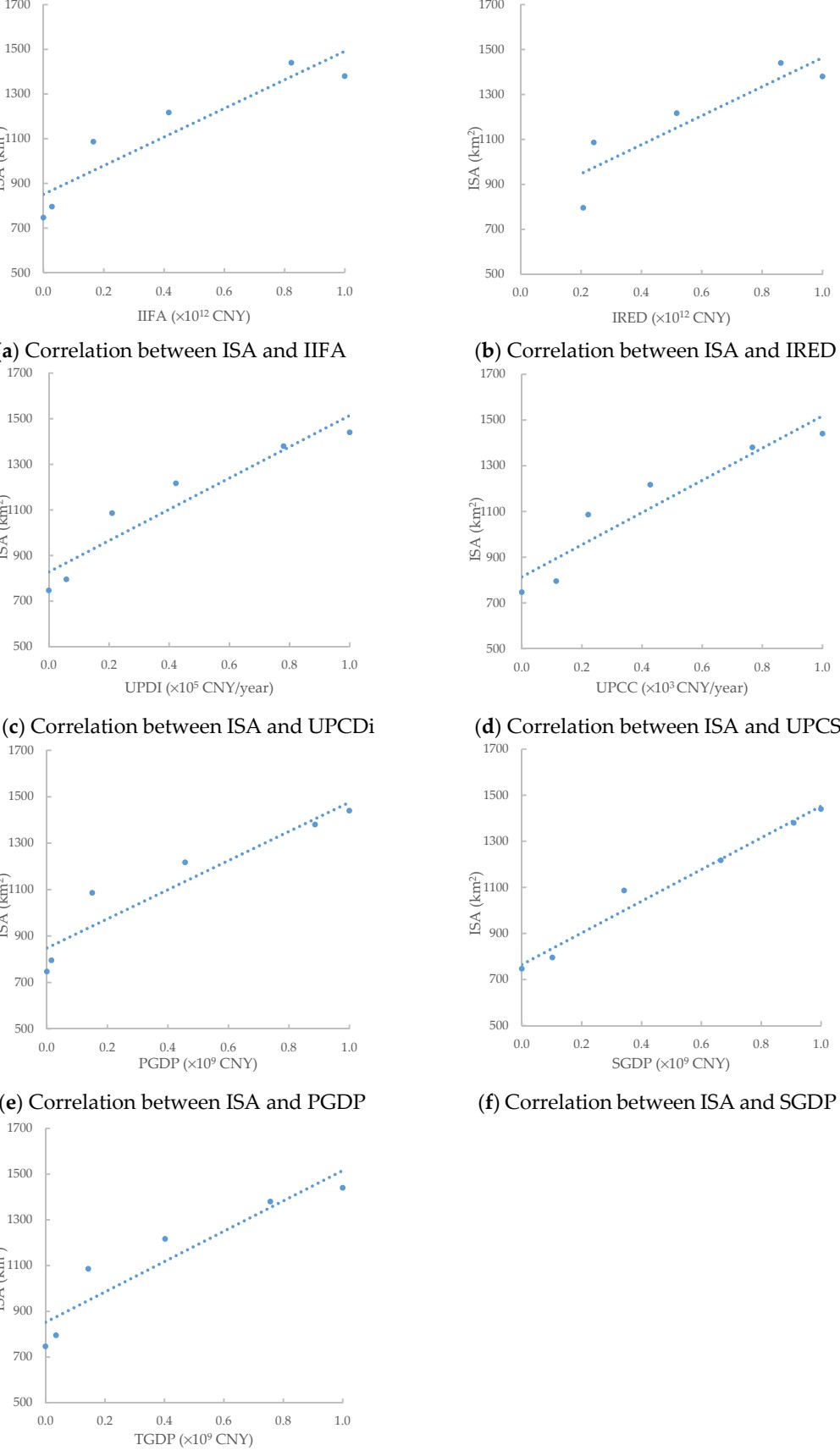

**Figure 11.** Correlation between ISA and socioeconomic indicators.

### 4.1.3. Cultural Factors on Urban Expansion

Urban culture determines the direction and quality of urban economic development. Dongying is the hometown of Sun Wu, a great ancient military strategist, and the birthplace of Lv Opera, a local representative opera in Shandong [66]. A great living environment, profound history and culture, and hospitable people are of great cultural attraction to urban migrants. Based on the Dongying yearbook, selecting from scientific, comprehensive, systematic, and other principles for factors, and in line with the actual situation of the city and the layout of cultural construction, the following four cultural indicators are selected as the driving factors: year-end population (YEP), cultural expenses (CE), cultural relic operation expenses (CROE), and inbound tourists (IT) (Table 7).

**Table 7.** Dongying cultural indicators from 1995 to 2018.

| Indicators | YEP ($\times 10^5$) | CE ($\times 10^5$ CNY) | CROE ($\times 10^5$ CNY) | IT ($\times 10^3$ Person Times) |
|---|---|---|---|---|
| 1995 | 16.411 | — | — | — |
| 2000 | 17.213 | 87.1 | 87.1 | 1.028 |
| 2005 | 18.05 | 216.46 | 216.46 | 1.3 |
| 2010 | 18.487 | 654.55 | 654.55 | 33 |
| 2015 | 19.062 | 1005.4 | 1239.4 | 58 |
| 2018 | 19.668 | 1648.6 | 1209.3 | 64 |

The four cultural indicators in Table 7 have increased at each indicator from 1995 to 2018. As illustrated in Figure 12, the four cultural indicators exhibited a positive correlation with ISA. The expansion trend of ISA is generally consistent with the growth trend of cultural indicators. Under the promotion of culture, urban economic development drives urban development. The unique urban culture promotes the development of Dongying city, and the pioneering and enterprising culture constructs the economic foundation of the development of Dongying city and adds vitality to the city.

### 4.2. Limitations of the Work

Although the proposed method has achieved satisfactory results in mapping the abundance of ISA, there were some limitations. First, high–quality Landsat remote sensing images of Dongying could not be obtained from 1995 to 2018. To overcome nonstationary and discontinuous spectral components in remote sensing image time series, a robust method of jump detection was proposed based on the anti–leakage least–squares spectral analysis (ALLSSA), along with an appropriate temporal segmentation, namely, Jumps Upon Spectrum and Trend (JUST) (https://doi.org/10.3390/rs12234001 (accessed on 7 July 2021)) [67], which can be applied to simulate vegetation time series with varying jump location and magnitude, the number of observations, seasonal component, and noises. Subsequent studies should be performed to determine the temporal and spatial variation of ISA by analyzing the time series of vegetation, as the area of vegetation is generally a non–ISA area. Second, although LSMA has been broadly considered as one of the most efficient methods, issues such as the endmember variability have not been considered. Because of the spatial heterogeneity of the landscape, the distribution of land use types differ for various regions, and the fixed end element type may lead to estimation error. Third, the spectral characteristics of high albedo ISAs and bare land are very similar, making it challenging to distinguish them. For Dongying, as a coastal city, it is difficult to separate the beach from the ISA. Therefore, higher image resolution and more advanced endmember extraction methods can improve the accuracy of ISA extraction. Fourth, the socioeconomic and cultural indicators used in this study account for the whole of Dongying City. Monitoring of the ISA at the urban scale reflects the overall trend in Dongying city expansion but ignores the different roles and interactions of different parts of the city in urbanization, leading to an insufficient understanding of the details of urban expansion. Using statistical data for districts and counties, and extracting the ISAs of each district and county, may improve the accuracy and specificity of the results.

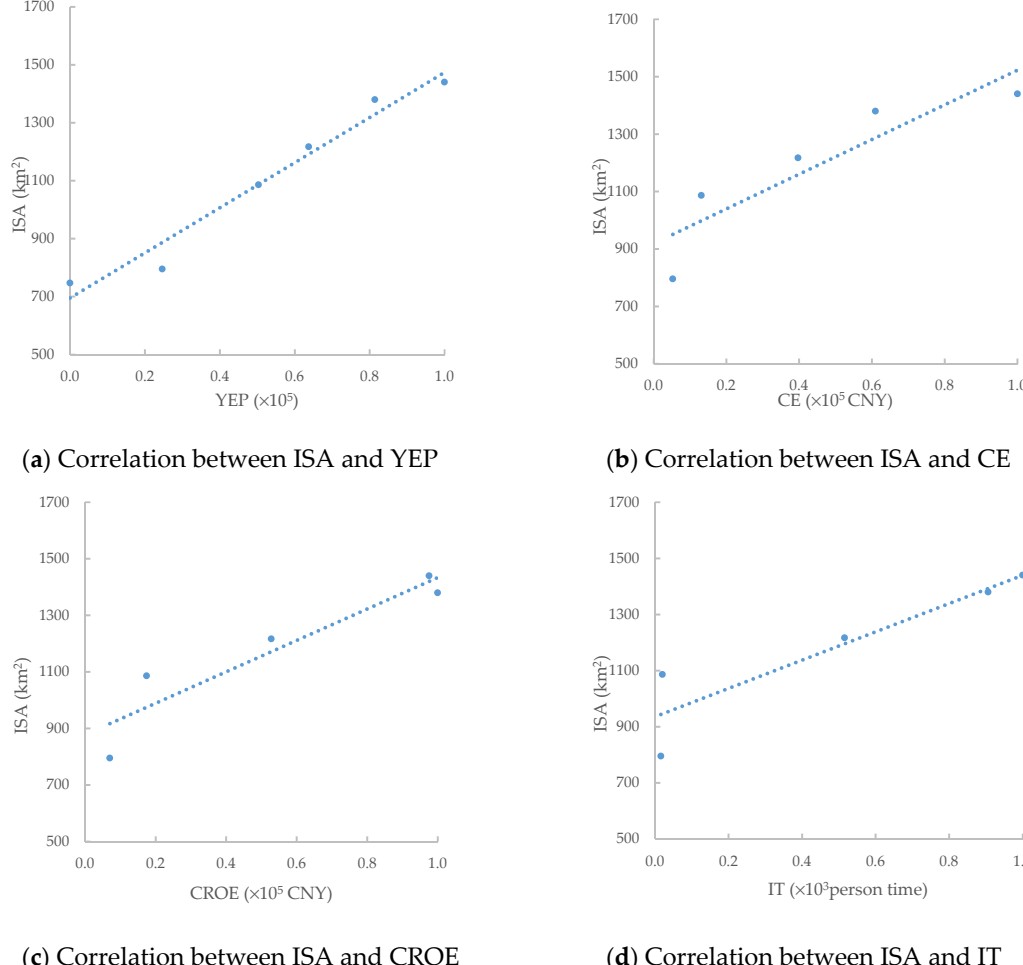

(**a**) Correlation between ISA and YEP    (**b**) Correlation between ISA and CE

(**c**) Correlation between ISA and CROE    (**d**) Correlation between ISA and IT

**Figure 12.** Correlation between ISA and cultural indicators.

## 5. Conclusions

Based on Landsat TM data of 1995, 2000, 2005, and 2010, as well as Landsat OLI data of 2015 and 2018, the ISA distribution, variation trends and potential driving forces over Dongying were investigated, With the following conclusions drew out.

(1) A pixel–based linear spectral mixture decomposition model was adopted for ISA extraction. The retrieved ISA thematic map fit the limited requirement of root mean square error (RMSE). Accuracy of ISA correct classification is greater than 83.08%. Further, the cross–comparison exhibits the general consistent with the ISA distribution of the land use classification map published by the National Basic Geographic Information Center.

(2) Research shows that the gradual increasing trend can be captured on the expansion of ISA from 1995 to 2018. The scope of ISA was the smallest in 1995, but it was large in 2015 and 2018. Despite of the central region always shown as the high ISA density, it still keeps increasing annually and radiating the surrounding region, especially in the southward which has formed into a new large–scale and high intensity of ISA in 2015–2018. Though the ISA patches scattered in the west region or along the northern and eastern part of the ocean coastline are still small, the expansion trend of ISA can be detected. The EII of ISA measuring the situation of its expansion changes from the lowest value 0.12% between 1995 and 2000 up to the highest 0.73% between 2000 and 2005. This shows that urban development was relatively slow from 1995 to 2000 and relatively fast from 2000 to 2005.

(3) Our investigation shows that expansion of ISA over Dongying is related to three driving forces (physical geography, socioeconomic factors, and urban culture). Richly

endowed by nature, the city's natural geographical environment provides an elevated chance of further urbanization. The rapid increase of regional economy provides a fundamental driving force for expanding ISAs. The development of urban culture promotes the sustainable development of ISAs. These three driving forces interact with the ISA and provide a good foundation and conditions for ISA expansion. Our findings provide a scientific basis for future urban land use, construction planning, and environmental protection in the Dongying area.

**Author Contributions:** Conceptualization, Y.S.; Methodology, J.S.; Validation, Y.C. and P.L.; Formal analysis, X.M.; Investigation, J.S.; Data preparation and organization, J.S. and Y.C.; Writing—original draft preparation, J.S.; Write—review and editing, P.L.; supervision, Y.S. and P.L.; Project administration, Y.S. and P.L.; Funding acquisition, Y.S. and P.L. All authors have read and agreed to the published version of the manuscript.

**Funding:** This study was supported by the National Key Research and Development Program of China (Grant No. 2020YFA0608501, Grant No. 2017YFB0504204), the National Natural Science Foundation of China (Grant No. 42071351), the Fundamental Research Funds for the Central Universities (Grant No. 2021YQDC01) and the Ecological–Smart Mines Joint Research Fund of the Natural Science Foundation of Hebei Province (Grant No. E2020402086), Liaoning Revitalization Talents Program (No. XLYC1802027), One Hundred Talents Program of CAS (No. Y938091), Project supported discipline innovation team of Liaoning Technical University (No. LNTU20TD–23), and Liaoning Key Program Serving for the Social–Economy Development of Towns at North–West Liaoning (No.10147–0816–1).

**Data Availability Statement:** All data are available from the corresponding author by request.

**Acknowledgments:** The authors would like to thank the open dataset of satellite images provided by USGS and the open survey data published by Chinese local government, and as well as Congying Shao, Jiapeng Huang, Yu Ma, Chongyang Wang and anonymous reviewers for their constructive comments and suggestions that improved the quality of this work.

**Conflicts of Interest:** The authors declare no conflict of interest.

## Abbreviations

**Symbol Table**

| | |
|---|---|
| $L_\lambda$ | measured spectral radiance |
| $DN$ | recorded electrical signal value |
| $G$ | sensor gain |
| $B$ | sensor bias |
| $L$ | total radiance received by the pixel at the sensor |
| $\rho$ | pixel surface reflectivity |
| $\rho_e$ | average surface reflectance around the pixel |
| $S$ | atmospheric spherical albedo |
| $L_a$ | atmospheric backscattering emissivity (atmospheric radiation) |
| $A$ | coefficient depending on atmospheric conditions |
| $B$ | coefficient depending on geometric conditions |
| MIR | mid-infrared band |
| Green | green light band |
| $b$ | number of spectral bands, $b = 1,2 \ldots 6$ |
| $x_b$ | reflectivity of band $b$ |
| $k$ | $k$th end element, $k = 1,2,3,4$ |
| $s_k$ | proportion of the area occupied by $k$ |
| $a_{kb}$ | reflectivity of $k$ in band $b$ |
| $e_b$ | residual |
| $n$ | total number of endmembers in the image |
| $S_a$ | ISA at the beginning of the study |

| | |
|---|---|
| $S_b$ | ISA at the end of the study |
| $T$ | time interval |
| *TLA* | total area of the study area |
| **Name Abbreviation Table** | |
| ISA | Impervious Surface Area |
| MNF | Minimum Noise Fraction |
| PPI | Pixel Purity Index |
| LSMA | Linear Spectral Mixture Analysis |
| USGS | the United States Geological Survey |
| NLCD | National Land Cover Database |
| VIS | Vegetable–Impervious surface–Soil |
| TM | Thematic Mapper |
| OLI | Operational Land Imager |
| DN | Digital Number |
| BIL | Band Interleaved by Line |
| FLAASH | Fast Line-of-sight Atmospheric Analysis of Spectral Hypercubes |
| MNDWI | Modified Normalized Difference Water Index |
| RMSE | Root Mean Square Error |
| IIFA | Investment in Fixed Assets |
| IRED | Investment in Real Estate Development |
| UPDI | Urban Per capita Disposable Income |
| UPCC | Urban Per Capita Consumption |
| PGDP | Primary industry Gross Domestic Product |
| SGDP | Secondary industry Gross Domestic Product |
| TGDP | Tertiary industry Gross Domestic Product |
| YEP | Year-End Population |
| CE | Cultural Expenses |
| CROE | Cultural Relic Operation Expenses |
| IT | Inbound Tourists |
| ALLSSA | the Anti-Leakage Least-Squares Spectral Analysis |
| JUST | Jumps Upon Spectrum and Trend |
| EII | Expansion Intensity Index |
| SMA | Spectral Mixture Analysis |

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
