# Peer review of "Extraction and Spatio-Temporal Analysis of Impervious Surfaces over Dongying Based on Landsat Data"

_remotesensing, doi:10.3390/rs13183666_

Round 1

Reviewer 1 Report

I would like to thank the authors for addressing the comments that I provided in my first report. The manuscript looks better now. I have a few more comments to be considered:

Lines 209-211. The authors should mention that positive values of MNDWI indicate water. Is this the threshold that the authors use?

Lines 285-290. I see the new Figure 6 shows a different spatial variation compared to the previous version. Did the authors make a mistake in their calculations before? If so, please make sure that you have updated the new information in the paper. I see no updates in lines 285-290? Please ensure that the correct information is provided in your new version according to the new results.

Figure 8. The caption needs more details. What is the unit of the x-axis? Why the x-axis values are so small?

The flowchart in Figure 3 appeared twice: on page 6 and on page 7. I assume that this is due to the highlighted changes.

Thank you for adding Figure 9 with the description in 386-392.

1) The references must be checked one by one to ensure their correctness and formats. First names should be initials, article volume, number, or page numbers must be consistent. Please follow the MDPI guidelines.

For example, references 13, 63, 64 should be respectively corrected as:

  1. Shen, Y.; Shen, H.; Cheng, Q.; Huang, L.; Zhang, L. Monitoring Three-Decade Expansion of China’s Major Cities Based on Satellite Remote Sensing Images. Remote Sens. 2020, 12, 491. https://doi.org/10.3390/rs12030491
  2. Ghaderpour, E.; Vujadinovic, T. Change Detection within Remotely Sensed Satellite Image Time Series via Spectral Analysis. Remote Sens. 2020, 12, 4001. https://doi.org/10.3390/rs12234001
  3. Shiru, M.S.; Shahid, S.; Alias, N.; Chung, E.-S. Trend Analysis of Droughts during Crop Growing Seasons of Nigeria. Sustainability 2018, 10, 871. https://doi.org/10.3390/su10030871

Also, references 57-60 and 62 are incomplete. More details should be added such as the URL and the last time the authors successfully had access to them. For example:

International Union of Pure and Applied Chemistry Home Page. Available online:
http://www.iupac.org/dhtml_home.html (accessed on 24 April 2005).

Line 421. Another style issue: please insert a dot after et al so it should be et al.   

Line 503. The link or reference to the paper describing the JUST software could also be added here, please see my first report.

2) The abbreviation table on page 28 needs attention. The first letters must be capitalized. For example, Expansion Intensity Index (EII) not expansion intensity index (EII). This formatting must be consistent throughout the entire paper and the abbreviation table. For example, line 98, should be read as Modified Normalized Difference Water Index (MNDWI). Please check the abbreviations one by one to resolve this issue.

Please carefully proofread the article before publication.

Thank you for your contribution

Regards,

Reviewer 2 Report

Summary 

The paper entitled “Extraction and spatio-temporal analysis of impervious surfaces in Dongying based on Landsat data” is a revision of previously submitted paper. In this paper, the authors presented their work of extracting impervious surface area (ISA) from Landsat images using Minimum Noise Fraction (MNF), pixel purity index (PPI), and linear spectral mixture analysis (LSMA). The authors did an adequate job on extracting ISA and analyzing its rate of change for their study period. The authors have also tested the correlation between three identified driving factors (physical geography, socioeconomic factors, and urban culture) and ISA expansion; however, the results were not consistent with the claims they made on the point 3 of the conclusion.   

I have outlined my comments as major comments and specific comments below:

Major comments 

Authors need to put effort in writing the manuscript in a clean and organized manner. Some of the sentences are not delivering the meaning they intended to. A paper with highlights and strikethrough all over was not an easy read.

I am not convinced with the details the authors have provided on the accuracy assessment part.

 Authors’ discussion on driving factors and conclusion no. 3 are not strongly founded on the results they have presented.

Specific comments 

 Line 54: Do not begin the paragraph with “therefore”. Move the paragraph up or put two closely connected statements on same paragraph.

Line 56-59: Need sentence revision

Line 61: “traditionally method” is incorrect use of the parts of speech; change it to “traditional”

Line 67: SMA appears at the first place; spell it out.  

Line 68: Isn’t that vegetation instead of vegetable on VIS?

Line 75-78: needs the citation number for Yang’s work on the first line it is mentioned.

Line 94: Don’t use the language such as “this paper understands”. You could write “provides insight” instead.

Line 127: Why do we need a citation for this line?

Figure 1: The font color they used for the text in top-right figure is not legible. They have to choose different color. Also, why did they use arrow-head line for the bottom-right figure to show study area? Is not the entire figure the study area?

Line 140: A calibration coefficient is then used. What does then mean here?

Line 147-150: What is the name of the module?

Line 170-174: Does the model need these parameters? If so, write that down.

Figure 2: There are two figures (top and bottom) and the caption does not mention about what these two figures are.

Figure 3: Why are there two exact same figures? The caption is so short; it does not describe what some acronyms (MNDWI) of the figure mean.

Figure 4: caption is not enough and does not say a word about what each of the two figures are.

 Line 209: can be got? Improve the language.

Line 228-235: Not sure, if this detail is needed here about MNF.

Line 255-267: needs a reference

Line 271: what does the word “extremum” mean?

Section 3.1: This section should not be in result section; instead, you could move it up to the preceding section.

Figure 6: Why am I seeing two exact same figures? Do colors on the legend and on the map match?

Figure 7: How was the RMSE calculated? What is the reference data to calculate RMSE?

Table 3: What is the reference data again?

Lines 386-392: What does it mean then? Write the implication in a sentence of using this image vs that image.

Figure 10: What do these numbers on the line represent? The highest slope and the largest difference between these numbers is between 2000 and 2005 as opposed to what authors have said in the paragraph that follow.

Line 421-423: I am not clear on what I am reading.

Section 4.1.1 Did the natural factors the authors mentioned were less favorable after 2005? Is that the reason the urban expansion was highest in 2000-2005 and then went down? They did not discuss these variations.

Section 4.1.2: If the ISA is so well correlated with each socioeconomic factors and increase rate of socioeconomic factors was significantly greater in 2010-2015, why the period does not match with the highest ISA expansion intensity index?

Section 4.1.3: The period of maximum increase of cultural factors and highest urban expansion, do not match here too.

Line 499-507: I was not clear whether it is a suggested method (ALLSSA) for the future use or was applied in this work to lessen the limitations.

Line 521-526: How is the study related to your study?

Conclusion no 3: I do not agree with the authors on this. Their data of maximum/minimum increase did not match.

Reviewer 3 Report

It seems that I have been invited to review a revised version of the manuscript! Anyway, this work is though does not offer any novelty, it does have many issues (outlined below) as it currently stands. First of all, English expressions have been so difficult to understand. Secondly, the intro part does not set out logically to understand motivation of the work. I would suggest to link with land use/land cover change induced impervious surface area increased. As such following works (https://www.sciencedirect.com/science/article/pii/S1674987121000888; https://www.nature.com/articles/nature01675; https://www.nature.com/articles/s41586-018-0411-9?source=Snapzu) could be highly useful. Regions experiencing high population growth tends to have larger ISA area – this is already known but what you want to prove in this work – this does not say clearly. You referred to previous words or statement in the work but in many cases, they are very unclear. See line 57: what are ‘those’? Line 169: absorption is not accounted for the sensors you used. Flowchart: instead of cutting, should be ‘sub-setting or clipping’. Line 208-214: Poor English, what do you mean by ‘this file’ in line 213? Line 288: What is ‘low…. Glass’? is there any high ref glass? Why sub-section 3.1 is in the result section, you did not develop this rather used an inbuilt tool from a software. If you wish to put this, send off to method section. Sub-section 3.3: I would not say this as you can’t define precision of ISA distribution? I would not use Table 3 in the MS send off to sup info. Line 376-377: Unclear statements. Line 440: What are the external factors? Show and analyse them here in the work to do a comparison between internal and external. What were the temporal resolution of ‘internal’ factors used? Nothing noted. I have strong reservation with Fig. 11: if you include only a few data points (in your case 6) you must have a BEST FIT curve always, this means use of large point could provide a useful outcome, though this was not the case.  Should be ‘Limitations of this/the work”.  I must ask you how lines 520-527 are relevant to your work? Drought is not part of your work so how did you link them? I suggest removing these.  

Round 2

Reviewer 3 Report

I have no further comments 

This manuscript is a resubmission of an earlier submission. The following is a list of the peer review reports and author responses from that submission.

Round 1

Reviewer 1 Report

General Comments:

This manuscript discussed the relationship between impervious surface area (ISA) distribution, variation trends, and their driving forces in Dongying, Shandong Province from Landsat images. This research article is technically excellent and suitable to publish in remote sensing. The objective of the article is clear, and the results are back up with proper images. I recommend publishing in current form.

Minor Comments:

  • Line 70, reference number should be 22.

Reviewer 2 Report

Reviewer’s report on the manuscript entitled:

Extraction and Spatio-temporal analysis of impervious surfaces in Dongying based on Landsat data

The authors used Landsat imagery to investigate the relationship between Impervious Surface Area (ISA) distribution, variation trends, and their driving forces in Dongying. They extract distribution, area changes, and expansion ratio via their proposed methodology and investigate their driving factors, such as geography and economy, and urban culture feature. My main concern is assessing the reliability of the model results because I see in Figure 6 that subregions tend to be non-homogeneous year after year. Is it due to the model error or is it due to human activity and/or climate change?

Section 2.2. Is the imagery only Landsat 5? Did authors also use Enhanced Thematic Mapper plus (ETM+) onboard Landsat 7 and 8? Please explain.

Line 30. Is 1 after stadiums a citation? Should be [1]?

The reference in line 40 should be [22].

Line 207. Please insert a space between transform and account

Equation (5), please put space between 0 and 1 and separate them by a comma.

Line 269, use lower case letter for variable x_b.

Line 270. k ranges from 1 to 4, and so the total number of endmembers in line 271 is n = 4. Is that correct?

Figure 6. I would prefer showing ISA in red and non-ISA in green! Because green usually is used to show vegetated areas!

Lines 284-290. Please also explain the changes from 1995 to 2000 and from 2000 to 2005. Why the red region in the east of the map in 2000 became green in 2005? Is it due to the model error (variations in principal components, etc.) or urbanization or climate change? Do you have any ground truth data for the assessment?

Although interesting, I think Figures 10 and 11 should just be used for extra information but not assessing the model! The authors should emphasize this in the Discussion.

The authors should select a few sub-regions in the map with known ground-truth data and investigate why ISA and non-ISA fluctuate within the subregion to assess the reliability of their results. I understand that the authors mentioned this as a limitation of their study in lines 418 to 421, but at least the selection of a few subregions will show the reader that the results presented in the paper are reliable.

Line 327-329. As an overall measure for the entire map yes but what about for subregions that tend to be non-homogeneous year after year as shown in Figure 6? The non-homogeneity could easily be because of a false model approach.

The images used are for January and February only as described in Section 2.2. Authors must also show the results for other months, at least the same process for July and August to see how the homogeneity of subregions change in a different season. This could also be another assessment strategy. Furthermore, I think vegetation indices, such as Normalized Difference Vegetation Index (NDVI)  and Enhanced Vegetation Index (EVI) can also be used for distinguishing between ISA and non-ISA regions and so as an assessment tool.

An abbreviation table is useful to be added to the end of the manuscript.

Finally, there are some recent trend and spectral analysis techniques that authors may want to consider for their data processing or at least include them in the Introduction or Discussion sections:

For example:

https://doi.org/10.3390/su10030871 or https://doi.org/10.3390/rs12234001 whose software is freely available at https://geodesy.noaa.gov/gps-toolbox/JUST.htm. For example, the software can be applied to Landsat image time series to show how ISA changes temporally and spatially via analyzing vegetation, water, and soil moisture time series.

Regards,

Reviewer 3 Report

Review Summary:

I enjoyed reading this paper entitled “Extraction and spatio-temporal analysis of impervious surfaces in Dongying based on Landsat data” in which the focus is to use Landsat data from five time periods in a quasi-regular interval to extract impervious surface area (ISA) and further analyze it in the Dongying city of China. The authors used linear spectral mixture analysis (LSMA) to extract ISA features for the months of January/February from the years between 1995-2018 and have found that ISA has grown between the period. The researchers used some preprocessing steps (radiation calibration), atmospheric correction, water masking, noise reduction and extract end-member pixels for major land cover categories found in the area before applying LSMA. The authors have used well-established models to come up with ISA growth between the years. The comparison between other relevant factors and ISA expansion is a unique aspect of this study. The authors have also attempted to seek if pleasant natural environmental factors, economic growth, and cultural abundance drove the urban expansion and hence, the ISA growth in the city. The paper is well-written and organized in a very easy-to-follow structure.

Since authors have not run proper statistical analysis and their comparative analysis is not between the data from spatially consistent units, the word “driver” is used in a more qualitative sense here. Even though many of those analyzed factors exhibit a linear correlation, the dependency is not well-grounded due to the lack of regression and confidence statistics between ISA and potential causative factors.  Also, besides the calculation of RMS, there is no factual data to verify how improved their result is because of NMF and PPI. These concerns should not be the major issues to turn down the paper, but the paper would have been footed on a solid foundation and would have offered much more if we saw those issues addressed.  

I have outlined my specific comments below:

Specific Comments:

Line 30: Is the number 1 at the end of the sentence citation?

Line 104: what is the difference between permanent population and population?

Line 105: rate of urbanization 69.24% per what time unit?

Line 105-107: Was the city of Dongying ranked no. 31? That is not clear from your sentence.

Figure 1: Does it have to be Landsat image when you are showing your study site? I think putting China’s national boundary in the inset map and using extent indicator for Dongying would serve the purpose better.

Line 133: Does the radiometric calibration module you mentioned have a name?

Line 138: Which software platform was used to run the FLAASH model? A sentence or two on how FLAASH works would be appropriate to add.

Figure 2: Put the legend to indicate the meaning of different colors (reflectance) on your image.

Line 158 and 159: Refer this sentence to subsections 2.3.2 and 2.3.4 so that the reader can expect to read more on MNF and PPI later in the paper.  

Figure 3: What is MNDWI water index? You spelled it out in the text that appears after this figure. So, in its first appearance, it is still unclear.

Figure 4: It also needs legend.

Line 194: Elaborate the meaning of “noise between spectral band”

Line 207: transformaccount needs word split

Line 230: greatest probability of what?

Line 237-239: “threshold factor” and “noise level” are the terms that need explanation for a reader unfamiliar to these

Line 249: low albedo materials are obviously low reflective materials. So instead of making it redundant writing that again, give more examples.

Line 250-251: you already masked the water, right?

Figure 6: the color used for non-ISA is misleading. For me, the 2005 image shows more non-ISA than the 2018.

Line 327-329: the concept of ISA expansion rate is still unclear. Put how you calculated it and what does rate indicate?

Line 386-387, Table 5: a short description on how these indices is derived would be helpful for readers.

Reviewer 4 Report

ine 30 What is "stadiums 1".

Line 42 Add citations. Just quote one.

Line 70 Citation error.

Line 114 Add as a reference?

Line 120, 153, 191, 223, etc. Some figure caption are poor in explanation.

Line 177, 269 Change the indentation of the text. Or put the first word in capital letters.

Line 262-263 That's more of Adams, et al. (1993).

Line 285-291 I do not get it. ISA increases and decreases successively. It does not correspond to the ISA and non-ISA in the Figure 6. And therefore it does not match the Table 2.

Line 377 One of the dots is missing in the Figure 10a.

In subsection "4. Discussion" there are no references for discussion. The ones used are superfluous. They are not scientific references.

The references are focused on China and some on the USA, and one on Australia. There are more studies elsewhere.